# Amino Acid Profiles in the Biological Fluids and Tumor Tissue of CRC Patients

**DOI:** 10.3390/cancers16010069

**Published:** 2023-12-22

**Authors:** Marisa Domingues Santos, Ivo Barros, Pedro Brandão, Lúcia Lacerda

**Affiliations:** 1Colorectal Unit, Hospital de Santo António, Centro Hospitalar Universitário de Santo António, 4050-651 Porto, Portugal; pedronunobrandao.cirurgia1@chporto.min-saude.pt; 2UMIB—Unit for Multidisciplinary Research in Biomedicine, ICBAS—School of Medicine and Biomedical Sciences, University of Porto, 4050-313 Porto, Portugal; ivo.silva.barros@gmail.com (I.B.); lucia.lacerda@chporto.min-saude.pt (L.L.); 3ITR—Laboratory for Integrative and Translational Research in Population Health, 4050-313 Porto, Portugal; 4Genetic Laboratory Service, Centro de Genética Médica Jacinto de Magalhães, Centro Hospitalar Universitário de Santo António, 4050-651 Porto, Portugal

**Keywords:** colorectal cancer, biomarkers, amino acids, metabolomics, diagnosis, prognosis, treatment response, laboratorial findings

## Abstract

**Simple Summary:**

Amino acids are the fundamental building blocks of proteins and play a crucial role in various cellular functions, such as protein metabolism/catabolism pathways and redox signaling. They are also vital in normal and cancer metabolism. The concentrations of amino acids in blood, urine, and tissue samples of cancer patients differ from those of healthy individuals. These differences can be used for cancer screening, monitoring treatment response, and predicting tumor prognosis. Colorectal cancer (CRC) is one of the most common and deadly cancers. This review aims to gather reported amino acid abnormalities in blood, urine, and tissue samples of CRC patients and how these abnormalities can be used as potential tools in the multifold management of CRC, including screening, establishing cancer prognosis, and monitoring treatment response.

**Abstract:**

Amino acids are the building blocks of proteins and essential players in pathways such as the citric acid and urea cycle, purine and pyrimidine biosynthesis, and redox cell signaling. Therefore, it is unsurprising that these molecules have a significant role in cancer metabolism and its metabolic plasticity. As one of the most prevalent malign diseases, colorectal cancer needs biomarkers for its early detection, prognostic, and prediction of response to therapy. However, the available biomarkers for this disease must be more powerful and present several drawbacks, such as high costs and complex laboratory procedures. Metabolomics has gathered substantial attention in the past two decades as a screening platform to study new metabolites, partly due to the development of techniques, such as mass spectrometry or liquid chromatography, which have become standard practice in diagnostic procedures for other diseases. Extensive metabolomic studies have been performed in colorectal cancer (CRC) patients in the past years, and several exciting results concerning amino acid metabolism have been found. This review aims to gather and present findings concerning alterations in the amino acid plasma pool of colorectal cancer patients.

## 1. Introduction

Colorectal cancer can be considered a severe public health problem [1]. The third most frequent cancer presents considerable mortality, motivated by a diagnosis often made at an advanced stage of the disease. On the other hand, whenever detected early, the probability of controlling the disease is high, with reasonable survival rates at five years [2]. Thus, combating it depends on effective screening programs, early diagnosis, and individualized therapies with continuous and effective monitoring. These are the only ways to reduce the incidence and improve the outcomes of treated cases regarding survival and quality of life, simultaneously reducing the economic burden.

Currently, control of this type of cancer is based on three aspects: screening, diagnosis, and treatment. In screening programs, fecal occult blood tests and colonoscopies (on a smaller scale) are still the most used methods starting from 50 years of age in asymptomatic individuals. However, implementing molecular tests that measure the immune system’s response to the attack of colorectal cancer in screening programs could prove to be an essential step to achieve. A colonoscopy beyond screening is the main weapon in prevention (a colonoscopy with polypectomy prevents patients with adenomatous polyps from developing colorectal cancer) and tumor diagnosis. When the diagnosis of invasive cancer is made, it is necessary to carry out correct staging to plan an individualized therapy that may include surgery, radiotherapy, and chemotherapy in different modalities. Monitoring the response to treatment, screening for recurrence, and selecting the ideal treatment are also essential aspects of managing this oncological entity.

Biomarkers are substances that can be measured in biological fluids or tissues and predict the incidence or outcome of disease [3]. A promising biomarker should fulfill five essential requirements: good specificity, good sensitivity, cost effectiveness, ease of implementation, and the ability to detect the disease at its early stages. While sensitivity refers to the ability of the biomarker to detect diseased patients, specificity relates to its capability to reject other pathologies or healthy individuals.

Colorectal cancer-specific biomarkers are still an area in development. Many biomarkers with different characteristics are currently being analyzed, which can be investigated in blood, urine, or tissue using various methods with different sensitivities, specificities, and clinical practice applicability. One example of these markers is the carcinoembryonic antigen (CEA) measured in blood, which is the most common tool used in the clinic for disease monitoring after resection surgery. Others include tumor biomarkers, such as Kirsten rat sarcoma virus (KRAS) and V-Raf murine sarcoma viral oncogene homolog B (BRAF) mutations or microsatellite instability, which have implications for therapeutic selection [4]. In the case of predictive biomarkers for response to neoadjuvant treatment in Locally Advanced Rectal Cancer (LARC), most biomarkers so far have been imaging variables and genetic parameters, as well as DNA- and RNA-based (TYMS polymorphisms, cell-free DNA, expression studies, microRNAs) [5,6,7]. Some are expensive, require state-of-the-art technology and complex laboratory procedures, and are not fully applied to the clinic yet.

Within this scope, whether in a targeted or untargeted fashion, metabolomics seems to be a tool for developing new biomarkers in cancer. Metabolomics uses body fluids and techniques to study low molecular weight molecules, such as amino acids, lipids, carbohydrates, and intermediates of various metabolic pathways. Some are already established procedures for screening and diagnosing other diseases, such as metabolic disorders. Metabolomics further allows the study of pathways known to be deregulated in cancer cells, providing a more thorough picture of the patient’s metabolic state. Glycolysis, the TriCarboxylic Acid Cycle (TCA), and mitochondrial metabolism have gathered much attention concerning cancer metabolism’s peculiarities. Their vast replicative potential, their metabolic plasticity, and their choice of aerobic glycolysis and concomitant lactate production as a source of energy—the Warburg effect—have made cancer cells an interesting source of possible metabolites for screening [8].

Among the vast array of molecules that might play a primordial role in the pathogenesis of cancer, amino acids have been attracting substantial attention. They have not only their role as building blocks for the synthesis of proteins but also can be considered alternative fuels in cancer due to their involvement in biosynthetic pathways [9]. For instance, glutamine and branched-chain amino acids provide organic molecules to fuel the TCA cycle; glutamate, glycine, and cysteine are used for glutathione synthesis, an antioxidant molecule essential to counter the production of reactive oxygen species during cell proliferation. In the same way, polyamines resulting from arginine catabolism have been associated with cancer development for decades [10]. Proteins involved in amino acid metabolism have also been under cancer research. They include transmembrane solute carriers and amino acid transporters, such as L-type amino acid transporter 1 and alanine–serine–cysteine transporter 2, which are usually upregulated in colorectal cancer [11]. Even though there is substantial biochemical evidence for the relevance of amino acids in cancer, there is still a lack of clinical output focusing exclusively on their use as biomarkers for the diagnosis, prognosis, and monitoring of therapeutic response in CRC (Figure 1).

This review aims to thoroughly examine the most recent findings concerning disturbed amino acid metabolism in colorectal cancer and possible metabolic pathways involved and how these may provide new screening, diagnostic, prognostic, staging, and predictive biomarkers for disease treatment.

## 2. Methodologies and Biological Samples for Metabolic Profiling

### 2.1. Types of Biological Matrices

Metabolic profiling in cancer is essential to discover shifts in the levels of specific metabolites in the cells and, concomitantly, disturbances in biochemical pathways and their underlying molecular mechanisms. These may be ultimately used for screening, staging, prognosis, and monitoring disease progression. The advantages of metabolomics come from the possibility of using minimally invasive procedures to collect biological materials such as blood, urine, or feces. Urine and blood are commonly used in metabolomics, although saliva, breath, cerebrospinal fluid, and feces are uncommon. Deciding which type of biological matrix best fits each study is a crucial step of any research in this field, as each type may fit a different purpose. For instance, amino acids in the blood can be analyzed in serum or plasma, providing a snapshot of an individual’s metabolism and revealing systemic changes provoked by disease. The serum is obtained by letting the blood rest at room temperature to coagulate fully, followed by centrifugation to precipitate cellular components. In contrast, for plasma, blood is collected in tubes containing an anticoagulant substance (heparin or EDTA) and then centrifuged immediately. The faster procedure required for the latter has the advantage of having a reduced risk of hemolysis, an undesirable process when analyzing amino acids in circulation. Additionally, plasma seems to yield better reproducibility, while serum tends to have a more significant concentration of specific metabolites, including amino acids [12,13]. For instance, glycine, arginine, phenylalanine, and serine have a higher concentration in serum relative to plasma, which may result from the coagulation process required for the analyses of this type of biological matrix [13].

Tumor tissue is organ-specific, allowing scientists to directly observe metabolic changes in cancer cells, as opposed to systemic variations in biological fluids that may not be as pronounced and may be affected by biochemical processes from the various tissues in the body. It requires, however, more complex and thorough processing procedures, such as using liquid nitrogen for immediate freezing and organic solvents for tissue homogenization and the extraction of metabolites. It also involves invasive collection procedures, such as a colonoscopy in the case of CRC, making it less suited for screening and diagnosis. Additionally, the amount of biopsy collected during a colonoscopy may not be enough for certain types of analysis. Only surgical removal of the tumor provides a considerable amount of tissue. For this reason, biomarkers applied in the clinic are usually assayed in biofluids, such as blood or urine. However, cancer tissue may be necessary to search for post-surgical prognostic biomarkers or help uncover pathways and mechanisms involved in developing the disease.

### 2.2. Common Methodologies Utilized in Metabolomics

The metabolome is usually studied using spectroscopic techniques, like nuclear magnetic resonance (NMR) or high-resolution magic angle spinning magnetic resonance spectroscopy (HR MAS MRS)), spectrometric techniques (mass spectrometry), and separation techniques, such as liquid and gas chromatography LC-MS and GC-MS. The choice of technique depends on several factors, such as the amount of biological material, targeted metabolites, concentration, type of metabolomic analysis (targeted or untargeted), budget, and others. Each methodology has its advantages and pitfalls. NMR has a high resolution, but it requires large quantities of material and is quite expensive and low in sensitivity, meaning that several metabolites may be below the detection limit and go undetected; LC-MS and GC-MS have high sensitivity a wide dynamic range, and are highly reproducible, but are still relatively expensive and high maintenance. It may be the case that more than one technique is required to cover a wide range of metabolites [14].

For the quantitative targeted analysis of amino acids, liquid chromatography currently remains one of the simplest and most used separation techniques [15]. More specifically, ion exchange liquid chromatography with ninhydrin post-column derivatization is a highly accurate and reproducible method for amino acid analysis in biological fluids such as serum, plasma, urine, and cerebrospinal fluid [15]. This technique is currently used in the clinic to diagnose and monitor inherited metabolic diseases, allowing the study of several amino acids [16]. Although it has a runtime of approximately 130 min, its lower costs are a significant advantage over mass spectrometry-related techniques, which are equally sensitive and offer superior selectivity, although they are more expensive and complex.

## 3. Amino Acid Profiling in Colorectal Cancer Patients

### 3.1. Cancer Tissue Metabolomics

Research has shown that transmembrane amino acid transporters are upregulated in colorectal cancer [11], facilitating their import to the cell. These transporters are now considered possible therapeutic targets. A larger concentration of amino acids in CRC tissues compared to normal mucosa is a hallmark in the published literature, which is consistent with a higher need for nutrients and the upregulation of protein turnover to support the high proliferation rates of cancer cells [17,18,19,20,21]. It has also been shown that this increase in amino acid content is also a distinctive feature between colorectal tumors and advanced adenoma, with satisfying sensitivity and specificity [18].

Another common trend seems to be an upregulation of glutamate in tumor tissues, which may be related to augmented glutamine degradation by cancer cells [19,22,23,24,25,26]. The prognostic potential of tissue glutamate has further been confirmed with a large sample of 376 surgical specimens from four independent cohorts of patients, using gas chromatography as an analytical platform [24]. Glycine also presents increased concentration in tumor specimens, with possible prognostic capabilities [18,20,27].

### 3.2. Serum and Plasma Metabolomics in Colorectal Cancer Patients

Although studies focusing exclusively on plasma-free or serum amino acid levels in colorectal cancer patients are scarce (Table 1), a common observation among published data is a decrease in the concentration of several amino acids in circulation [28,29,30,31,32], such as tyrosine, methionine, and threonine. Therefore, most studies reviewed here have used a more holistic approach, analyzing a full metabolic profile of CRC patients and not only a specific class of metabolites.

#### 3.2.1. Glutamine and Glutamate Metabolism

Glutamine is the most abundant free amino acid and an important player in cancer proliferation due to its role as an alternative fuel and anaplerotic substrate of the TCA cycle [33]. Even being a non-essential amino acid, “glutamine addiction” is a known hallmark of cancer cells. As such, it is worth investigating if significant disturbances in its metabolism are found in circulation in CRC patients. In serum and plasma, glutamine seems to be consistently downregulated in CRC patients [6,31,34,35,36], which may be related to the excessive consumption of this amino acid by the tumors. Only one study verified the opposite trend [28].

There are inconsistencies in what concerns serum glutamate. While some authors describe an elevation of circulating glutamate in CRC patients compared to controls [35,36], others have seen the opposite trend [6,28,37]. It is worth mentioning that the work by Tan et al. is remarkably consistent due to the use of a patient-set validation in a large test population [6]. Furthermore, using two different chromatography techniques, these authors could distinguish stage I patients from healthy controls.

#### 3.2.2. Glycine and Serine Metabolism

Glycine is a component of glutathione and is one of the main antioxidant molecules of the cell, and, as such, it has a relevant role in redox homeostasis [38]. An upregulation of glycine and serine biosynthesis is a characteristic of cancer cells, which produce intermediates to synthesize these amino acids through their high rates of aerobic glycolysis. Nonetheless, there is no clear trend regarding circulating levels of these amino acids in CRC, as both upregulations and downregulations have been observed for glycine [30,35] and serine [6,30,35].

#### 3.2.3. Tryptophan Metabolism

Tryptophan, an essential amino acid, is known to have a role in intestinal health—it is catabolized by the gut microbiota to produce indoles, compounds involved in numerous aspects of gut homeostasis. The kynurenine pathway, through which most of the ingested tryptophan is catabolized, generates compounds that induce the proliferation arrest of T lymphocytes and, as such, suppress the antitumor immune response [39]. In this regard, tryptophan has been correlated with immune activation and quality of life in CRC patients [40], and targeting its catabolism is currently sprouting several lines of investigation. Data obtained using several analytical platforms have shown a clear tendency of reduced circulating tryptophan in CRC patients [6,31,36,37,41], including a study using plasma from early-stage volunteers (0/I/II) [42,43]. Interestingly, pre-operative serum tryptophan has been singled out as a possible discriminator between colon and rectal cancer [44].

#### 3.2.4. Branched-Chain Amino Acid Metabolism

These essential amino acids, which include valine, leucine, and isoleucine, are used by tumors for biosynthetic purposes. By activating the mammalian Target Of Rapamycin (mTOR) pathway, they can stimulate protein translation and are a nitrogen source for nucleotide synthesis. They are also incorporated into the TCA cycle through the production of Acetyl-CoA (acetyl coenzyme A). Interestingly, blood tests for the detection and recurrence of CRC based on their metabolism have been reported [38]. Diminished circulating levels of branched-chain amino acids in CRC patients seem quite common [28,29,31,45,46,47]. Geijsen and colleagues verified this phenomenon for valine and leucine in CRC plasma using 268 patients and 353 controls [45]. Serum isoleucine has also been reported as a possible discriminator between colon and rectal cancer [44].

#### 3.2.5. Proline Metabolism

Proline is used in collagen synthesis, the main component of the extracellular matrix (ECM), and has roles in cell adhesion and migration and the development of tissues. Both depleted [30,35,37] and raised [20,36] levels of serum proline have been found in CRC patients compared to healthy individuals.
cancers-16-00069-t001_Table 1Table 1Studies that related amino acid levels in serum/plasma in patients with colorectal tumors compared with healthy individuals (amino acids with diagnosis/screening potential).ReferenceYearType of CancerTNM StagePatients (*n*)Healthy Controls (*n*)Biological SampleLaboratory TechniquesTan et al. [6]2013CRCI/II (68%)101102SerumGC-TOFMS/UPLC-QTOFMSChan et al. [20]2009CRCVarious stages3131SerumHR-MAS NMR/GC-MSBarberini et al. [29]2019CRCVarious stages159PlasmaGC-MSLeichtle et al. [30]2012CRCVarious stages5958SerumET-MSTroisi et al. [34]2022Adenomas, CRCI/II15050SerumGC-MSGu et al. [35]2019Adenomas, CRCVarious stages6238SerumGC-MSZhu et al. [37]2014Adenomas, CRCVarious stages14292SerumLC-TMSNishiumi et al. [42]2017Adenomas, CRCI/II282291PlasmaGC-T/QMSNishiumi et al. [43]2012CRCI/II5963SerumGC-MSGeijsen et al. [45]2019CRCVarious stages268353PlasmaUHPLC-QTOF-MSFarshidfar et al. [46]2016Adenomas, CRCVarious stages320254SerumGC-MSMa et al. [47]2012CRCVarious stages3020SerumGC-MSWang et al. [48]2017CRCI/II5540UrineH-NMR**Reference****Glutamate****Glutamate****Glycine****Serine****Threonine****Tryptophane****Proline****Valine****Leucine****Isoleucine**Tan et al. [6] DD
I
D



Chan et al. [20]





I


Barberini et al. [29]






DDDLeichtle et al. [30]

I


D


Troisi et al. [34]D








Gu et al. [35]DIDD

DDDDZhu et al. [37]DI


DI


Nishiumi et al. [42]




D



Nishiumi et al. [43]DI







Geijsen et al. [45]






DD
Farshidfar et al. [46]






DDDMa et al. [47]






DDDWang et al. [48]

DDDD



Legend: I—increase; D—decrease; GC-MS—gas chromatography–mass spectrometry; UHPLC-QTOF-MSa—ultrahigh performance liquid chromatography–quadrupole time-of-flight mass spectrometry; H-NMR—proton nuclear magnetic resonance spectroscopy; ET-MS—electrospray tandem–mass spectrometry; HR-MAS NMR—high-resolution magic angle spinning nuclear magnetic resonance; GC-TOFMS—gas chromatography–time-of-flight mass spectrometry; UPLC-QTOFMS—ultraperformance liquid chromatography–quadrupole time-of-flight mass spectrometry; GC-T/QMS—gas chromatography/triple–quadrupole mass spectrometry.


## 4. Amino Acids as Discriminators of Colorectal Cancer Stage

An important question in cancer metabolomics is whether more pronounced anomalies are associated with increased TNM stage (the staging system includes T for tumor, N for node, and M for metastasis) in circulating metabolites. If so, these changes could be used to differentiate cancer stages. Unfortunately, studies focusing on metabolic biomarkers able to discriminate CRC stage are scarce, and although serum metabolomics has unveiled interesting findings, these are yet to have true staging capacity. Until now, the most significant differences in circulating amino acids are associated with stage IV disease. Notably, a large international consortium of 744 CRC patients from four cohorts has revealed significant decreases in plasma citrulline and histidine in patients with stage IV compared to patients with stage I [49]. Other scientific articles have also found decreased serum histidine in metastatic CRC compared to locoregional disease [42,50]. However, other works have used a considerably smaller number of samples, rendering the results less accurate [51,52].

## 5. Amino Acids as Prognostic Biomarkers in Colorectal Cancer

Elevated glycine concentrations in tumor tissue (>1.77 µmol/g) have been associated with worse progression-free survival in LARC patients, although no association with response to neoadjuvant therapy has been made [53]. As reported earlier in this review, glutamate is commonly upregulated in tumor tissue and has also been found to have prognostic potential [24].

For circulating biomarkers, serum glutamine has been singled out for its prognostic potential, as two research groups have identified lower serum glutamine as being an adverse prognostic factor in CRC, as well as having an association with pro-inflammatory cytokines [54,55] (Table 2). Further investigation of this issue has also revealed that decreased pre-operative serum glutamine is correlated with shorter overall and disease-free survival [56]. Other authors have identified serum amino acids with prognostic potential in metastatic disease and recurring stage II patients, although with no clear tendency [46,57].

## 6. Amino Acids as Predictive Biomarkers of Response to Neoadjuvant Therapy

LARC comprises T3–4 or node-positive (T1–4N+) tumors. Treatment usually consists of neoadjuvant chemoradiotherapy (nCRT) with 5-FluoroUracil (5-FU) or capecitabine to help sensitize radiation, followed by resection surgery. The assessment of the response to treatment can be based on tumor regression grades (TRGs). While various systems exist, the Mandard TRG system has been proven efficient in previous research [58,59]. Predicting response to neoadjuvant chemoradiotherapy is crucial for improving cancer therapy since it would allow personalized treatment for patients who do not benefit from it. Due to their involvement in cancer metabolism, amino acids may be potential biomarkers for response prediction in LARC patients.

Only four publications have identified amino acids with the predictive power of response to neoadjuvant therapy (Table 3). So far, published data have been obtained using different biological matrices, methodologies, neoadjuvant regimens, TRG scales, and a low number of samples, which may lead to different outcomes. In addition, the authors used different time points, including a different collection of biological material before neoadjuvant treatment. Still, research has shed some light on the predictive powers of some amino acids. These included the identification of branched-chain amino acids as possible predictors of response—valine with a reported 80% sensitivity and 41% specificity [28] and leucine with 71% sensitivity and 83% specificity [60]—as well as metabolites from glycine and histidine metabolism [61]. Moreover, differences in concentration in plasma metabolites from tryptophan catabolism between responsive and non-responsive patients also reinforce the interest in this metabolic pathway. However, no predictive biomarkers have been implemented [62].

## 7. Usefulness, Applicability, and Limitations of Metabolic Abnormalities Analysis in Cancer in General and CRC in Particular

In the last twenty years, metabolomics has tried to look for biomarkers in colorectal cancer, recurring to techniques such as chromatography, nuclear magnetic resonance, and mass spectrometry. Considering the current lack of options with this potential in the clinic, the aim has been to find new molecules that could be used for screening, diagnostic, prognostic, staging, and predictive purposes. Due to their involvement in numerous metabolic pathways of cancer metabolism, amino acids are good candidates for this role. They are also readily detectable using methodologies currently in practice to screen other diseases.

Analyzing these literature subjects, it becomes clear that amino acid metabolism is affected by the onset of cancer, as patients have a distinct profile compared to healthy individuals. However, several discrepancies exist among published data due to the lack of a standardized protocol for analyzing these metabolites. Those studies often use different analytical methodologies, biological matrices, and test populations. The lack of a more significant sample number and one with a considerable number of patients with CRC early-stage (I/II) patients makes the studies published have a weak impact. The ideal screening biomarker should be able to detect the disease prematurely. However, among other works reviewed here, the most significant percentage of patients with CRC early-stage (I/II) patients has been 68 percent [6]. So far, only two publications have addressed patients solely with CRC stage I/II [43,48].

Another interesting point not usually approached is whether changes in tumor tissue are reflected in plasma or serum. One publication has shown commonly dysregulated pathways between tissue and pre-treatment serum, such as alanine, aspartate, and glutamate metabolism (upregulated) and glycine, serine, and threonine metabolism (downregulated) [41]. However, cysteine and methionine metabolism seem to have an opposite trend between tissue and circulation. Still, the overall conclusion is that the degree of dysregulation seems to be more pronounced in cancer tissues than plasma. However, there is still a lack of knowledge regarding this subject.

Glutamine metabolism seems to be the most affected pathway across the several publications reviewed here. As much of the glucose is metabolized to lactate even in aerobic conditions (Warburg effect), glutamine acquires an important role as an anaplerotic substrate and has long been known to be a source of addiction for cancer cells [63]. For these reasons, glutamine metabolism has acquired substantial interest among the scientific community as a cancer therapeutic target [33]. Altered glutamine and glutamate levels in the serum of CRC patients may result from cancer cells’ dependence on glutamine. Lower concentrations of plasma-free glutamine have also been reported in other cancers, such as lung [31,64] and breast cancer [65], which further reinforces the potential of this amino acid as a diagnostic biomarker.

Stage TMN is currently the most effective prognostic factor in CRC, with more advanced stages associated with worse survival. Carcinoembryonic antigen (CEA) is commonly used for postoperative follow-up but has limited accuracy concerning its prognostic potential. Other clinical prognostic markers nowadays include KRAS and BRAF gene mutations and microsatellite instability. Pre-treatment high tumor glycine seems to have an adverse effect on the progression-free survival (PFS) of LARC patients [53]. The authors further observed that this variable was superior to the T stage as a predictor of poor PFS in multivariate regression analysis. Similarly, high serum glycine has a deleterious effect on the cancer-specific survival of CRC [55]. Significant associations between high serum glycine and worse overall survival have also been observed in head and neck cancer [66].

Glycine is an amino acid used for glutathione synthesis—an antioxidant molecule that can prevent cellular damage from agents, such as reactive oxygen species. So, higher glycine concentrations in tumors may help cancer cells protect against reactive oxygen species and keep their redox balance, rendering tumors more aggressive. This way, glycine consumption has also been associated with the stimulated proliferation of cancer cells [67].

Similarly, low serum glutamine is a biomarker of poor prognostic in CRC [54,55], which is consistent with tumor consumption of this amino acid by cancer cells. The number of publications using serum metabolomics for distinguishing CRC stages is still scarce [49,51,52], and only stage IV patients have presented interesting findings, such as depleted histidine and citrulline, compared to earlier stages. Vahabi et al. not only used a small group of sixteen patients but also discriminated stages roughly into two groups (stages 0–I and II–IV) [51].

Metastatic cancers are an overall different disease—the tumor has spread to other organs, which may elicit more pronounced anomalies in the metabolic profile of the patients. These may be due to modifications in tumor biology or a difference in the body’s response to the disease. There is still much work to be performed to find biomarkers that are able to discriminate cancer stages, particularly early disease, such as stages I and II, and correlate these with survival. However, stage IV biomarkers may be good candidates for patient follow-up after resection surgery, ideally detecting even small occult metastasis.

Four scientific papers using serum- or plasma-based metabolomics to discover predictive biomarkers of response to neoadjuvant therapy were reviewed here. These have all used different methodologies and sample sizes, and the highest number of patients enrolled was 105 [61]. Also, the neoadjuvant regimens have not been the same, nor have the regression grades used in response evaluation, which may lead to discrepancies between publications. Branched-chained amino acids have been acknowledged as possible discriminators between groups of response to therapy. These are known to stimulate tumor growth by inducing biosynthetic pathways through the activation of the Target Of Rapamycin Complex 1 (TORC1) and by fueling the TCA cycle through conversion to acetyl-CoA [9]. In fact, enzymes involved in their catabolism are known to be overexpressed in several carcinomas [68].

Increased plasma tryptophan and kynurenine have also been identified in non-responsive LARC patients compared to responsive ones [62]. Tryptophan has gathered considerable attention due to its immunomodulatory role. In contrast, kynurenine, which results from tryptophan catabolism by tryptophan 2,3-dioxygenase (TDO) and indoleamine 2,3dioxygenase (IDO), is known to favor a tumor’s immune escape by inducing CD8 T-cell death [69] and is elevated in several cancers [9]. Indoles produced from the catabolism of tryptophan by gut bacteria also have a significant role in intestinal homeostasis [70].

In cervical cancer patients, tryptophan and valine have also been selected as biomarker candidates for the prediction of response to neoadjuvant chemotherapy [71]. As detected by NMR, serum threonine, glutamine, and isoleucine are possible discriminators of the type of pathological response in breast cancer [72]. Similarly, three significantly disturbed metabolic pathways in triple-negative breast cancer patients have been identified: glycine, serine, and threonine metabolism; valine, leucine, and isoleucine biosynthesis; and alanine, aspartate, and glutamine metabolism. Significative differences in branched-chain amino acid biosynthesis were mainly observed when comparing stable disease, complete response, and partial response groups [73].

### Highlights

Amino acid abnormalities can be detected in serum, plasma, and tumor tissue.Amino acid abnormalities in tumor tissue are more pronounced than in circulation; however, the methodology is invasive and more complex.To study amino acids as biomarkers in CRC, plasma or serum collection is less invasive and preferred.Ion exchange chromatography with ninhydrin post-column derivatization is a simple and effective method to analyze plasma-free amino acids. Standardized protocols used to diagnose metabolic diseases can be employed to study amino acids in cancer patients’ circulation.Circulating levels of glutamine, branched-chain amino acids, and tryptophan have the potential to aid in CRC diagnosis, as they are potential hallmarks of CRC patients.Some studies also indicate pre-operative serum tryptophan as a possible discriminator between colon and rectal cancer.Amino acid plasma levels can be used as prognostic factors in colorectal cancer. Notably, histidine and citrulline levels differ in stage IV CRC. High-tumor glycine and low-serum glutamine are adverse prognostic factors in CRC.In LARC patients treated with neoadjuvant therapy, serum tryptophan and branched-chain amino acid levels predict the response to CRT.

## 8. Future Perspectives

Although there is a considerable amount of output on this subject, clinically approved amino acids for screening, diagnosis, and prognostic are yet to be implemented. Biomarkers need sensitivity and specificity close to 100% to be feasible and need previous validation with many patients. Most of the works reviewed here had a relatively small group of patients. They lacked validation cohorts relevant to testing the discovered biomarkers in a new patient set to determine the results’ reliability and reproducibility [74]. Biomarkers without prior external validation will most likely never be used in a clinical setting.

One of the difficulties concerning metabolomics is its large amount of output. Although relevant to understanding cancer’s pathogenesis, new types of analysis should be considered and implemented. Machine learning (ML) has become an interesting alternative in various fields [34]. As a branch of artificial intelligence (AI), it encompasses a set of algorithms that can use pre-existing training data to “learn” a specific task, which can then be applied to new information. For example, in spectroscopic techniques, such as NMR and MS, deep learning, a branch of ML using artificial neural networks, has previously been used for pre-processing and analyzing data, peak identification, and integration [75].

Computational models that detect metabolic patterns representative of cancer onset are currently being studied by Huang et al. [76]. Also, in the last few years, machine learning has been used for the prediction of response to neoadjuvant therapy in breast cancer [77], lung cancer [78], and colorectal cancer [34,79]. As such, to consolidate the alterations in amino acids and other metabolites, metabolic machine learning models are an interesting next step for the future of screening, prognosis, and response prediction of colorectal cancer.

## 9. Conclusions

Amino acid plasma/serum levels are potential colorectal cancer biomarkers. They can be used in diagnosis, prognosis, and tumor response to therapeutics in colorectal cancer. The clinical implementation of these biomarkers depends on the definition of a laboratory test that is easy to implement, reproductive, effective, and economical; the realization of multicentric studies with the same design with more extensive data.

Artificial intelligence is also a promising methodology that can help data interpretation.

## Figures and Tables

**Figure 1 cancers-16-00069-f001:**
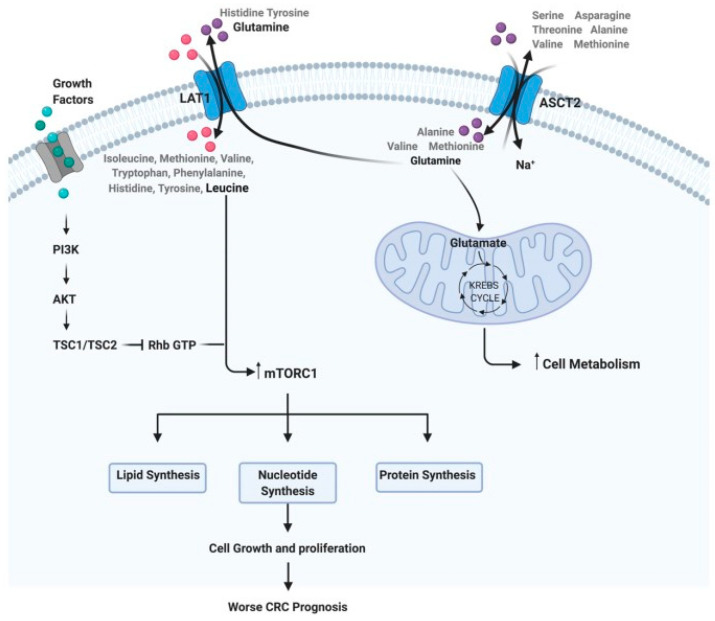
Steps of the amino acid system. For example: how glutamine and leucine levels can interfere with cell metabolism, growth, and proliferation, implying a worse prognosis in CRC (reprinted with permission—Dias et al., 2021 [11]).

**Table 2 cancers-16-00069-t002:** Studies that related amino acid levels in tumor tissue or serum with prognosis.

Reference	Year	Disease	Stage	Early Stage	Patients (*n*)	Healthy Controls (*n*)	Biological Sample	Techniques	Findings
Qiu et al. [37]	2014	CRC	TNM 0–IV	46%	227	Na	Tumor biopsies	Gas chromatography–time-of-flight mass spectrometry	Glutamate, aspartate, cysteine, and beta-alanine can distinguish patients with a longer time to recurrence and better 5-year recurrence.
Farshidfar et al. [52]	2016	CRC	TNM I–IV	33%	320	254	Serum	Gas chromatography–mass spectrometry	A model with 35 metabolites, including ornithine, proline, and aspartate, is able to discriminate recurring from non-recurring stage II patients.
Redalen et al. [53]	2016	LARC	T3–4/N0/M0 or T2-4/N1-2M0-	na	54	Na	Tumor biopsies	High-resolution magic angle spinning magnetic resonance spectroscopy	High tumor glycine (1.77 μmol/g) is associated with worse progression-free survival.
Ling et al. [54]	2019	CRC	TNM I–IV	49%	123	Na	Serum	Enzy ChromTM Glutamine Assay Kit	Lower plasma glutamine (<52 ng/μL) is associated with worse overall survival and progression-free survival.
Sirnio et al. [55]	2018	CRC	TNM I–IV	51%	357	Na	Serum	Nuclear magnetic resonance	Lower glutamine (<410 μmol/L) and histidine (55 μmol/L) and higher phenylalanine (>93 μmol/L) and glycine (>263 μmol/L) are associated with decreased cancer-specific survival.
Bertini et al. [57]	2012	mCRC	IV	na	181	139	Serum	Nuclear magnetic resonance	Lower serum creatine and valine are associated with shorter overall survival.

**Table 3 cancers-16-00069-t003:** Studies that related amino acid levels in plasma/serum with response to nCRT.

Reference	Year	Type Ofcancer	Stages	Patients (*n*)	Healthy Controls (*n*)	Biological Sample	Techniques	Findings
Rodríguez-Tomàs et al. [28]	2021	LARC-	TNM T3–4 and/or N+	32	48	Plasma	GC-EI-QTOF-MS	Significantly lower plasma valine in pre-nCRT samples from pathological complete response patients.
Yang et al. [60]	2018	Colon and Rectum	TNM II–III	47 Rectum; 10 colon	na	Plasma before nCRT	UHPLC—quadruple time-of-flight)/mass spectrometry analyses	Leucine increased in non-responsive patients with superior sensitivity to CAE and CA 199Jia.
Jia et al. [61]	2018	LARC	TNM T3–4 and/or N+	105	na	Serum	Liquid chromatography–mass spectrometry	3-methylhistidine, 4-imidazoleacetic acid, and dimethylglycine downregulated in nCRT-resistant patientsRodrı.
Crotti et al. [62]	2020	LARC	TNM T3–4 and/or N+	45	na	Plasma	UHPLC-UV-VIS/FLD and LC-MS/MS	Significantly increased tryptophan in non-responsive patients.

UHPLC-QTOF-MSa—ultrahigh performance liquid chromatography–quadrupole time-of-flight mass spectrometry. UPLC-UV-VIS/FDL—ultrahigh-performance liquid chromatography with ultraviolet and fluorescence detection. GC-EI-QTOF-GC/MS—gas chromatography–cold electron ionization–quadrupole time-of-flight–mass spectrometry. LC-MS/MS—liquid chromatography–mass spectrometry.

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
