# Peer review of "Amino Acid Profiles in the Biological Fluids and Tumor Tissue of CRC Patients"

_cancers, 2023, doi:10.3390/cancers16010069_

Round 1
Reviewer 1 Report
Comments and Suggestions for Authors
The article of Marisa Domingues Santos et al. entitled Amino acids profile in the biological fluids and tumor tissue of CRC patients is a review on determining the profile of basic amino acids as potential biomarkers for the early detection of colorectal cancer (CRC). This is a current topic because medicine still does not have appropriate biomarkers indicating the advancement of the disease, determining the prognosis of survival or the body's response to treatment of CRC. Amino acids, apart from their role as building blocks, can also act as fuel for a developing cancer, therefore, their level in body fluids could indicate the stage of cancer development. This review shows that little research has been conducted in this area. The studies described in this paper are often based on different methodologies and therefore it is difficult to draw clear conclusions. It is clear that this topic still requires a lot of research. This article is well written and pleasant to read. The description is very detailed and supported by the latest scientific literature. The 46 references (out of 79) come from the last decade, most of them from the last few years. I only noticed a few minor editing errors.
1. The manuscript lacks authors.
2. Chapter 4 has no content or maybe the remaining chapters should be subchapters (4.1, 4.2 and so on)?
3. In the line 310 there is a symbol „%”, but in my opinion there should be a word „percent”.
4. I found some minor typing errors, for example, in the line:
· 88 there is „figura 1” instead „figure 1”;
· 90 there is „aminoacid” instead „amino acid”;
· 134 there are two brackets;
· 167/168 and 195/196 and 418/419 the text jumps to the second line;
5. In References items 17, 60 and 82 are not articles.
Author Response
Thank you for the kind comments. The suggested changes were made.

Reviewer 2 Report
Comments and Suggestions for Authors
The manuscript could be potentially very interesting because it deals with the important topic concerning amino acids metabolism and colorectal cancer. However, several problems in the organization of the manuscript are present:
1. Section 3 discussed the alterations of amino acids in colorectal cancer. A summary table is recommended for better understanding.
2. Section 8 is similar to Section 4-7. These parts should be reorganized in order to highlight the theme.
3. Highlights in section 8 should be clearer and more concise.
Comments on the Quality of English LanguageCareful proofreading is recommended.
Author Response
Thank you for the comments made. I hope the changes made met your guidelines. In this sense, three tables were introduced. Section 8 was retained, although the title was changed. The aim is to frame the metabolic changes in CRC from the perspective of changes present in cancer in general, emphasizing all the difficulties in moving from research to clinical practice and the possible processes to overcome them. The highlights section has been shortened to make it more precise and concise.

Reviewer 3 Report
Comments and Suggestions for Authors
The review by Marisa Domingues Santos et al., is a good review that sheds light on important and promising markers for CRC (amino acids and their metabolites). Obviously, there is not enough research done so far, hence those markers are not in use yet. Hopefully, this review will encourage research groups in this field to carry out more investigational work. I have minor comments:
Please write the full name for BRAF and KRAS mutations
Acetyl-CoA (acetyl coenzyme A)
TNM staging system stands for Tumour, Node, Metastasis
Author Response

(The authors gave the same response as above.)

Round 2
Reviewer 2 Report
Comments and Suggestions for Authors
The paper can be accepted without any further changes.